# Systems thinking and complexity science methods and the policy process in non-communicable disease prevention: a systematic scoping review protocol

Chloe Clifford Astbury ![ORCID] ,[1] Elizabeth McGill ![ORCID] ,[2] Matt Egan,[3] Tarra L Penney[1]

[1]Global Food System and Policy Research, School of Global Health, Faculty of Health, York University, Toronto, Ontario, Canada
[2]Deaprtment of Health Services Research and Policy, London School of Hygiene & Tropical Medicine, London, UK
[3]Department of Public Health, Environments and Society, London School of Hygiene & Tropical Medicine, London, UK

**Correspondence to**
Dr Tarra L Penney;
tpenney@yorku.ca

## ABSTRACT

**Introduction** Given the complex causal origins of many non-communicable diseases (NCDs), and the complex landscapes in which policies designed to tackle them are made and unfold, the need for systems thinking and complexity science (STCS) in developing effective policy solutions has been emphasised. While numerous methods informed by STCS have been applied to the policy process in NCD prevention, these applications have not been systematically catalogued. The aim of this scoping review is to identify existing applications of methods informed by STCS to the policy process for NCD prevention, documenting which domains of the policy process they have been applied to.

**Methods and analysis** A systematic scoping review methodology will be used. Identification: We will search Medline, SCOPUS, Embase and Web of Science using search terms combining STCS, NCD prevention and the policy process. All records published in English will be eligible for inclusion, regardless of study design. Selection: We will screen titles and abstracts and extract data according to published guidelines for scoping reviews. In order to determine the quality of the included studies, we will use the approach developed by Dixon-Woods *et al*, excluding studies identified as fatally flawed, and determining the credibility and contribution of included studies. Synthesis: We will identify relevant studies, summarising key data from each study and mapping applications of methods informed by STCS to different parts of the policy process. Review findings will provide a useful reference for policy-makers, outlining which domains of the policy process different methods have been applied to.

**Ethics and dissemination** Formal ethical approval is not required, as the study does not involve primary data collection. The findings of this study will be disseminated through a peer-reviewed publication, presentations and summaries for key stakeholders.

## INTRODUCTION

Given the complex and inter-related causes of many non-communicable diseases (NCDs), and the complex realities in which policies designed to tackle them are made and unfold,

### Strengths and limitations of this study

► This scoping review protocol outlines the first piece of work to systematically identify and review how methods informed by systems thinking and complexity science (STCS) have been applied to non-communicable disease prevention policy.
► We use the Preferred Reporting Items for Systematic Reviews and Meta-Analyses Protocol checklist 2015 in reporting the systematic identification, screening, eligibility of included literature.
► This study will search journals from multiple disciplines to provide a more comprehensive picture of how STCS methods have been applied.
► This scoping review may miss studies that do not self-identify or use the language of methods informed by STCS.

the need for a 'system-level' approach to NCD prevention which encompasses complexity is increasingly recognised.[1] Systems thinking and complexity science (STCS) represent a multidisciplinary field of established and emergent theories and methods,[2] which may be applied to NCD prevention. While a contrast has been drawn between STCS as distinct traditions,[3] STCS approaches and methods are broadly characterised by the idea that real-world phenomena exist within systems composed of dynamic actors, including people, organisations and other structures, which evolve in response to each other and their contexts.[2]

### What role can STCS play in public health and health policy?

Methods informed by STCS have been applied to various phenomena of interest by a range of disciplines, and health researchers have explored their utility in solving seemingly intractable public health issues. These applications in public health are growing rapidly,

with as many as 90% of published examples appearing in the past decade.[4] Several reviews have documented existing approaches to applying STCS to methods and practice in public health. These reviews have made a number of contributions to clarifying terminology and theoretical framing of STCS in public health, including developing frameworks to assess and strengthen complex systems for chronic disease prevention[5] and outlining the range of STCS ideas referred to be public health researchers.[2] However, most reviews have commented on the relative paucity of studies documenting practical applications of such methods.[2 6–8]

Theoretical discussions around how STCS can be usefully applied to understanding and facilitating the policy process have highlighted the existence of a complex 'policy-making system', where networks of individuals, organisations and interests interact to produce emergent systemic behaviours.[9 10] Characteristics of complexity can be identified throughout the policy process: policy decisions are difficult to predict using deterministic models; policy decisions, once taken, may be implemented in dynamic ways adapted to local contexts by different actors; and implemented policies may have different impacts in different contexts.[9] However, discussions of STCS and the policy process have also questioned the extent to which STCS presents policy-makers with a 'new' way of approaching their work, given that policy-makers may already operate with an implicit awareness of the notion of 'complexity', independently of STCS theory.[10]

However, STCS-informed methods may have the added value of bringing more robustness to how policy-makers engage with the complexity of the policy process, and providing more opportunities to incorporate research evidence. Despite the emphasis on evidence-based policy in public health,[11] the role of research evidence in policy-making remains relatively limited,[12] with policy-makers often differing with researchers around what sort of evidence is 'good' and 'useful'.[13] Further, evidence generated by researchers may only be inputted at particular points in the policy process, with many parts of this process being a dynamic series of negotiations between different perspectives and interests. Given that policy-makers already operate in a continually evolving space, methods informed by STCS applied to different domains of their work may bring greater rigour and transparency to the process, and ultimately, the utilisation of evidence to inform policy. While many working in policy have expressed an interest in the promise of STCS methods to enhance policy, a recent study of policy evaluators concluded that the methods were in limited use, and that the pragmatic framing of these methods should be seen as a priority to ensure their greater penetration into the process.[14]

While examples of practical applications are limited, studies have demonstrated the benefits of applying methods informed by STCS to the health policy domain. A 2015 review of system dynamics modelling in support of health policy at any level of government reported that the method's key strengths included facilitating consensus building among stakeholders and providing policy-makers with dynamic, targeted tools to inform their decisions.[6] This review also highlighted ways forward for system dynamics modelling in health policy, including more user-friendly software; better communication of the advantages of system dynamics modelling to policy-makers; building capacity to enable more widespread use of this type of modelling; and generating evaluative evidence to illustrate the benefits of the method.[6] A 2019 review of system dynamics and agent-based modelling in mental health research, while identifying a limited number of empirical examples, commented on the potential applications of such methods to mental health policy, stating that they might be particularly useful in two processes: first, modelling the impacts of 'distal' policies, where the policy was removed from its potential impacts in terms of time or causality; and second, assessing what conditions might be necessary for a policy to be successful.[7] Finally, Johnston *et al* developed an STCS-informed framework which they used to assess a number of North American obesity policy documents.[15] This framework used the concept of 'leverage points' within systems, which identify different system components as having the potential to create more or less substantial change.[16] This analysis highlighted that many recommendations made in obesity policy focus on leverage points with limited potential to provoke substantial, system-level change.[15]

## Aims and scope

While there has been substantive discussion and theoretical development relating to STCS in the policy process, well-documented examples of how STCS approaches can be applied in policy-making for NCD prevention are less common. STCS methods may be usefully applied in other areas of public health characterised by complex interactions between multiple stakeholders and domains, as well as other disciplines more broadly. However, policy-making for NCD prevention has characteristics which may make STCS methods particularly useful. First, there are commercial actors and interests involved in NCD-related policy, including the tobacco, alcohol, and food and beverage industries. This adds additional complexity to the policy process and makes a case for transparent approaches to incorporating multiple perspectives and forms of evidence in making policy decisions. Second, despite concerted policy efforts to reduce the burden of NCDs, in many contexts progress has been limited, suggesting that a novel approach that encompasses complexity may be useful.[17 18]

Although some relevant examples of STCS methods in the policy process have been identified, these either do not result from a comprehensive and systematic review of the literature,[4 19] or are restricted to a specific method.[6] Other systematic reviews of STCS in public health do not focus specifically on policy-making,[2 5 8] which is a specific context and process in which STCS-informed methods

may have particular uses and important strengths and limitations.

Further, a gap exists in determining which of these methods are useful and practical for practitioners with different needs and levels of resource, and in making these distinctions accessible to potential users. Scholars of complex systems have previously emphasised the importance of increasing the use of methods informed by STCS in public policy processes, and the responsibility held by researchers to effectively translate their knowledge and methods to encourage their adoption in the policy process.[19 20] A review of existing practice which documents clear examples of how these methods can be applied in this context, as well as under what conditions a certain approach might be most useful, is an important part of this process of translation.

While STCS-informed approaches to understanding policy-making have emphasised its non-linearity, linear or cyclical models of the policy process remain in frequent use by policy-makers and practitioners.[21 22] In order to facilitate the practical use of review findings, we will use a cyclical model of the policy process (developed by the Centers for Disease Control and Prevention (CDC)[21]) to structure the process and results of this review. After Howlett and Cashore, we characterise the policy process as one which moves from broader 'goals' to concrete 'means': specific, on-the ground policy measures designed to achieve the stated goals.[23] We use the definition of the domains of the policy process developed by the CDC (see table 1).[21]

Further, for this review to be useful to its intended audience, it is important that it takes into consideration the resources available to policy-makers in generating evidence, as well as the ways in which evidence is applied in the policy process. Ghaffarzadegan *et al* highlight that methods used in the policy process must not only lend themselves well to insight generation, but also to being communicated, as decisions in the policy world must often be justified to stakeholders.[19] They argue that some STCS-informed methods might lend themselves more easily to this communication process than others, such as group model building, which supports diverse stakeholders in reaching a consensus, or a small systems dynamics model limited to a smaller number of components, making it easier to interpret.[19]

Therefore, the objective of this review will be to systematically identify and summarise existing applications of STCS-informed methods in NCD prevention policy, documenting key methodological elements and identifying which domains of the policy process they have been applied to.

## METHODS AND ANALYSIS

We will conduct a systematic scoping review of peer-reviewed literature documenting the application of methods informed by STCS to the policy process in NCD prevention. The scoping review will be conducted in line with guidelines published by Arksey and O'Malley and refined by Levac *et al*,[24–26] which emphasise an iterative approach suited to an exploratory research question.

In line with these guidelines, this review will be conducted in the following domains[24]:
1. Identifying the research question.
2. Identifying relevant studies.
3. Study selection.
4. Charting the data.
5. Collating, summarising and reporting the results.

### Stage 1: identifying the research question

Informed by our study objective, our central research questions are:
1. How have methods informed by STCS been applied in the policy process in NCD prevention? Which domains of the policy process and areas of NCD prevention policy have methods informed by STCS been applied to?
2. What practical considerations, such as advantages, limitations, barriers and facilitators, have been described in applying STCS-informed methods to NCD prevention policy?

By policy we refer to public policy, defined as 'a set of interrelated decisions taken by a political actor or group of actors concerning the selection of goals and the means of achieving them within a specified situation where those decisions should, in principle, be within the power of those actors to achieve'.[27] We understand policy as being ultimately in the hands of government, although we recognise that a number of limitations constrain the policy options available to government, including other domestic and international actors.[23 27] For the purposes of this review, we extend the definition of government to include supranational governing bodies.

**Table 1** The domains of the policy process, from oerview of CDC's policy process[21]

| Domain | Description |
| --- | --- |
| Problem identification | Clarify and frame the problem or issue in terms of the effect on population health |
| Policy analysis | Identify different policy options to address the problem/issue and use quantitative and qualitative methods to evaluate policy options to determine the most effective, efficient and feasible option |
| Strategy and policy development | Identify the strategy for getting the policy adopted and how the policy will operate |
| Policy enactment | Follow internal or external procedures for getting policy enacted or passed |
| Policy implementation | Translate the enacted policy into action, monitor uptake and ensure full implementation |

CDC, Centers for Disease Control and Prevention.

**Table 2** Concepts from the research question used in developing the search strategy according to the PCC framework

| Population | Whole population approach to NCD prevention |
| --- | --- |
| Concept | Methods and approaches informed by systems thinking and complexity science |
| Context | Policy-making and different domains of the policy process at different levels of government, including local, national and supranational |

NCD, non-communicable disease; PCC, Population Concept Context.

## Stage 2: identifying relevant studies

We will systematically search electronic databases for peer-reviewed literature (Medline, Scopus, Web of Science, EMBASE). This review will focus on peer-reviewed literature in order to identify the range of specific and distinct methods that are in use. As a result, applications of methods informed by STCS which are documented in the grey literature will not be identified.

The search strategy will be informed by the main concepts in our research question using the Population Concept Context framework recommended by the Joanna Briggs Institute for use in scoping reviews[28] (see table 2; see online supplemental file 1 for detailed search strategy). Search strategies will be developed iteratively, informed by existing systematic reviews focused on related concepts[5 6 29–31] and indicator papers meeting inclusion criteria of which the authors are aware. As initial searches generated numerous records relating to genetics (due to the inclusion of the term 'regulation' along with health-related terms), a block of NOT terms will be added to the search strategy.

## Stage 3: study selection

Records identified through the searches will be collated and double screened using the online platform Covidence.[32] Studies will be included where they meet all of the following criteria:

1. Primary study from any country or region, available in English.
2. Self-identify as taking an approach informed by STCS.
3. Report empirical findings from a piece of research done during a specific point in the policy process (problem identification, policy analysis, strategy and policy development, policy enactment, policy implementation, evaluation, stakeholder engagement and education).
4. Focus on a subject related to NCD prevention.

For academic records, titles and abstracts will initially be screened, followed by full-text screening. Full-text screening will be undertaken by two independent researchers, one of whom will have extensive experience in the area of STCS and NCD prevention (TLP or ETM, who have previously authored reviews on related topics[29 33]).

In order to facilitate the identification of methods which may not have been identified as STCS methods by previous reviews of the public health literature,[2 8 29] but which authors have identified as STCS methods, title–abstract screening will take an inclusive approach, and full texts will be screened to identify STCS language used to describe methods. Papers which focus on healthcare or clinical services rather than primary prevention will be excluded. Papers which concern potential risk factors for NCDs but focus on non-NCD outcomes (such as alcohol consumption as a risk factor for road traffic accidents or inter-personal violence) will also be excluded.

In line with published guidelines, the approach to study selection may be refined iteratively when reviewing articles for inclusion.[24–26]

## Stage 4: charting the data

Data charting will be conducted using a data charting form designed to identify the information required to answer the research question and subresearch questions (see table 3). As recommended, the data charting form will be piloted with five to ten records to ensure that it is consistent with the research question, and the data charting form will be revised iteratively in order to ensure the purpose of the research is being met.[24–26] Where the required information is not included in a report, we will follow up with named contacts for additional information.

## Stage 5: collating, summarising and reporting the results

We will undertake quality assessment of the including studies using the approach developed by Dixon-Woods *et al*, excluding studies identified as 'fatally flawed' in the first instance, and determining the credibility and contribution of included studies as part of the synthesis of the evidence.[34]

We will analyse the extracted data, presenting a numerical summary of the included studies in table form, allowing us to identify intersections between STCS methods, domains of the policy process and areas of NCD prevention policy. We will also conduct a thematic analysis of the contents of the included articles in order to identify, if possible, what needs these methods have met and the resources they require, and what challenges were encountered in applying the methods.

## Patient and public involvement

Patients or the public were not involved in the design, or conduct, or reporting, or dissemination plans of our research.

## STRENGTHS AND WEAKNESSES OF THE STUDY

This review will only identify examples of methods which have previously been applied in the policy domain, and where this application has been documented. We hope this will increase the value of our findings for practitioners, but as a result, methods that have not been

| Table 3 | Data charting form |
|---|---|
| Record | Title |
| | Authors/organisation |
| | Year |
| Application | Policy process (problem identification, policy analysis, strategy and policy development, policy enactment, policy implementation)[36] |
| | Rationale for using STCS (if stated) |
| | Area of NCD prevention (health outcome or risk factor) |
| | Policy level (local, national, regional, global) |
| | Stakeholders involved, if any (government, academic, professional, industry, community) |
| | Project (state if publication was part of a larger project) |
| | World region |
| Method | Name |
| | Tool used (if any: software, kit) |
| | Aim/research question (if stated) |

NCD, non-communicable disease; STCS, systems thinking and complexity science.

applied, or only applied in other fields, will not be identified in this review.

Further, studies that do not 'self-define' as using methods informed by STCS will not be included. Anzola *et al* highlight the existence of 'analogical' uses of terms relating to complexity, where central characteristics of STCS are employed or implicitly referred to without being explicitly linked to the relevant theory and methods.[35] Narrative reviews have previously identified implicit complexity concepts in the policy literature.[9] However, in the absence of shared terminology, such usage may be difficult to systematically identify in the literature given the reliance of the systematic review method on identifying key words and phrases. In order to include as many relevant examples as possible, we will conduct title–abstract screening in an inclusive way, progressing records to full-text screening if there is any uncertainty. Further, our search strategy has been designed to be relatively inclusive, including broad terms related to STCS, such as complexity and system lens or perspective, as well as specific methods previously identified in systematic reviews.

## ETHICS AND DISSEMINATION

Formal ethical approval is not required, as the study does not involve primary data collection.

The further involvement of methods informed by STCS in the policy process will support policy-makers in developing evidence-based solutions to complex problems that arise in tackling NCDs. This scoping review will identify existing applications of methods informed by STCS to the policy process. Review findings will provide a useful reference for policy-makers, outlining which domains of the policy process different methods have been applied to, and highlighting the resources they require and the problems they have addressed. The findings of this study will be disseminated through a peer-reviewed publication, presentations and summaries for key stakeholders.

**Acknowledgements** The authors would like to acknowledge the useful comments and contributions to the scope of the proposed review by members of the WHO European Office for Prevention and Control of Noncommunicable Diseases.

**Contributors** CCA and TLP conceived and designed the study. CCA drafted the manuscript. TLP, EM and ME provided critical input on the manuscript and methods, and read and approved the final manuscript.

**Funding** CCA and TLP acknowledge internal research support from York University, Toronto, Canada and the WHO European Office for Prevention and Control of Noncommunicable Diseases. EM and ME were supported by the National Institute for Health Research (NIHR) School for Public Health Research (SPHR), Grant Reference Number PD-SPH-2015.

**Disclaimer** The views expressed are those of the author(s) and not necessarily those of the NIHR or the Department of Health and Social Care.

**Competing interests** None declared.

**Patient consent for publication** Not required.

**Provenance and peer review** Not commissioned; externally peer reviewed.

**ORCID iDs**
Chloe Clifford Astbury http://orcid.org/0000-0003-2955-7833
Elizabeth McGill http://orcid.org/0000-0002-3841-8467

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
