## [Reviewer comments · BMJ Open]

ARTICLE DETAILS

TITLE (PROVISIONAL)	Systems thinking and complexity science methods and the policy process in non-communicable disease prevention: a systematic scoping review protocol
AUTHORS	Clifford Astbury, Chloe; McGill, Elizabeth; Egan, Matt; Penney, Tarra

VERSION 1 – REVIEW

REVIEWER	Abdool Karim, Safura Witwatersrand University Press
REVIEW RETURNED	24-May-2021

GENERAL COMMENTS	The protocol is a useful one and the manuscript succeeds in outlining a gap in the literature which this review seeks to fill. The section beginning at page 3 line 18 onwards requires further refinement to explain the utility of the STCS in public health policy - at present my view is that the two limited examples discussed at page 4 lines 8 and 14 are not explained in sufficient detail and are in themselves not a sufficient evidence based to justify the study. In addition, though the authors mention other reviews have been conducted, it would be worthwhile to expand on the general findings and more clearly articulate how this review is distinct. At a methodological level, it is not entirely clear why the review focuses on application to NCD prevention policies and whether there is sufficient literature to warrant a review. There are particular reasons why the policy making process for NCD prevention policies are distinct from other public health policies so there can be a justification for this scope but this is not clearly articulated in the current manuscript. Finally, it would be helpful for the authors to include some indication of the impact of the the limitation of including only studies that self-define as STCS and what that definition may look like. The authors state that they "hope this will not result in the exclusion of a large number of relevant studies" but do not indicate steps to ameliorate this risk should the criteria end up excluding a large number of studies.
---

REVIEWER	Rigby, Benjamin Durham University
REVIEW RETURNED	30-May-2021

GENERAL COMMENTS	Peer review
-------------

This scoping review protocol presents an interesting proposed project that will identify and review how methods informed by systems thinking and complexity science (STCS) have been applied to different aspects of the policy process in relation to non-communicable disease (NCD) prevention. This will give rise to numerous important considerations for policy-makers that align with this journal's publication themes. Discourse around STCS is increasingly prevalent, yet the application of these methods in relation to public policy are, as yet, not well known. With the likely increased burden of NCDs as a result of SARS-CoV-2 pandemic, and growing recognition of the need for STCS-based policy responses, this paper is both important and timely. It thus represents a valuable contribution to the field of STCS and policy.

Particular strengths of the scoping review protocol are that it highlights some of the practical issues associated with STCS implementation, as part of the rationale. The authors propose an appropriate and well defined scoping review methodology. The PRISMA-P extension has been used to ensure all relevant aspects of the protocol have been considered. The paper has been proof-read well. The title and abstract reflect the main body of the paper, which itself is of good length.

However, I have particular comments about the absence of certain literatures in the introduction, the structure of the manuscript, and the definition of key concepts and research questions. Therefore, I recommend relatively minor revision. To aid the authors in revising this paper, I offer more detailed comments as follows:

Substantive comments

Abstract

A couple of minor points:

1. Page 1 (line 18): Consider stating 'scoping' review in the aim of the study.
2. Page 2 (lines 7-8): 'This scoping review...' This reads like the aim of the study and thus duplicates your statement at the end of the introduction section of the abstract. Please remove this sentence and amalgamate with the previous statement.

Introduction

The manuscript will benefit from additional consideration of research that has previously explored applications of STCS, particularly theoretically, in the context of policy processes. I have provided some references below that the authors may wish to consider to help inform these considerations.

3. Page 3 (lines 7-9): Please reference this statement (*i.e.* STCS may be of value for those with limited time...)
4. Page 4-5 (Applying STCS to the policy process): When reading this section, I was searching for a definition of the policy process. I found it in the methods section (page 7, line 21; table 1). Please bring this definition and table forward or summarise some key elements of policy processes in your introduction to aid the reader. It would be beneficial to state that this is just one conception of the policy process, and that while some policy scholars may denounce such linear/cyclical

models, they are familiar to, and commonly considered by, policy actors.

5. Page 4-5 (Applying STCS to the policy process): This section would also benefit from an acknowledgement of work that has looked at how theories of STCS may map on to aspects of the policy process, even if just to say these matters have been considered more theoretically, and less so in practical terms. In particular, please see Robert Geyer's (Lancaster University) or more pertinently Paul Cairney's (Stirling University), work, which offers insights on how complexity theories may inform policy-making and policy-making processes, taking into account policy-makers' perspectives and working practices.

Methods and analysis

6. Page 7 (line 10): The second sub-research question is not clear. Do you mean to ascertain which methods mandate particular needs and levels of resource for policy-makers to use? Alternatively, do you mean which methods have been applied by policymakers according to their needs and the availability of resources to them? Furthermore, if the latter of these options, how do you intend to ascertain the level of resources policy-makers had access to; what sorts of resources will you consider charting from the data you retrieve (financial, human capital *etc.*)? Please clarify.
7. Page 7 (lines 11-18): Again, these useful definitions may aid the reader if included earlier in the introduction instead. However, I am happy to leave this to the author's discretion.
8. Page 8: (line 3): How did you develop your search strings? Please make reference to said search strings and the Appendix in this subsection of the methods.
9. Page 9 (lines 6 and 11): The focus of your scoping review is clear (*i.e.* STCS and NCD). However, what limits are you putting in place (*i.e.* do you have a pre-established set of STCS approaches/ NCDs you will consider)? I acknowledge STCS will be self-identified. However, this will then be subjective decision on the reviewers' part to include or exclude. While I am familiar with the authors' previous work in the field, others may not be. Please include a brief statement about the reviewers' requisite experience for undertaking this work. It will also be beneficial to offer a statement and/or examples of the kinds of STCS approaches and NCDs you expect to find data on.
10. Page 11 (line 5): This short statement alludes to Arksey and O'Malley's optional consultation phase of scoping reviews. Consider changing the subtitle to reflect this stage, as set out by Arksey and O'Malley, and briefly justify why you decided not to undertake it, particularly alongside policy-makers themselves.

Strengths and weaknesses:

11. Page 10 (lines 20-22): I was previously searching for a statement about why you were not including grey literature in the 'study selection' section on page 8. I do not see consider

this much of a limitation as, a few key government reports aside (e.g. Magenta Book 2020), I do not anticipate much grey literature on the topic of interest. I recommend moving this statement to the methods section.

Ethics and dissemination

12. Ethics and dissemination are, largely, separate considerations. I recommend using two separate subheadings, both on pages 11-12 and in the abstract.

Appendix

13. Please number your Appendix (i.e. Appendix 1). Check the publication guidelines; this may need to be termed and referenced 'supplementary material/file' both here and in the main manuscript.

PRISMA-P 2015 Checklist

14. Please double-check page numbers. There seem to be discrepancies between those stated in the checklist and where the information is found in the manuscript (e.g. funding information is on page 13, while the checklist says page 16).

General comments and reflections

15. Please be aware your page numbering seems to go wrong from page 10 onward (however, this may be a proof-formatting error).

Reference is made to Anzola's taxonomy of terminology associated with STCS. The authors may choose to reflect on this in the write-up of their review, to consider how policy-makers themselves discuss such concepts and thus may likely engage with the research findings.

Please be advised that new PRISMA reporting standards were published this year, and I recommend the authors reflect on these in their write-up.

Concluding remarks

I believe this scoping review will make a valuable contribution to the field of STCS and policy. I encourage the authors to consider and respond to my suggested revisions above. I offer the following references that may assist in this regard:

Cairney. (2012). Complexity theory in political science and public policy. *Political Studies Review*. **10**(3), pp.346-358.

Cairney. (2015). How can policy theory have an impact on policy-making? The role of theory-led academic-practitioner discussions. *Teaching Public Administration*. **33**(1), pp.22-39.

	Cairney and Geyer. (2017). A critical discussion of complexity theory: how does 'complexity thinking' improve our understanding of politics and policymaking? Complexity, Governance and Networks. 3(2), pp.1-11. Nobles et al. 2021. The action scales model: a conceptual tool to identify key points for action within complex adaptive systems. Perspectives in Public Health. doi:10.1177/17579139211006747. Page et al. 2021. The PRISMA 2020 statement: an updated guideline for reporting systematic reviews. BMJ. 372(n71). Weible et al. eds. 2017. Theories of the Policy Process. 4th ed. London: Routledge.
--	--

VERSION 1 – AUTHOR RESPONSE

Reviewer: 1

Dr. Safura Abdool Karim, Witwatersrand University Press

Comments to the Author:

The protocol is a useful one and the manuscript succeeds in outlining a gap in the literature which this review seeks to fill.

The section beginning at page 3 line 18 onwards requires further refinement to explain the utility of the STCS in public health policy - at present my view is that the two limited examples discussed at page 4 lines 8 and 14 are not explained in sufficient detail and are in themselves not a sufficient evidence based to justify the study.

RESPONSE:

Thank you for this comment. We have overall expanded the introduction section, adding more detail around reviews and commentaries and referencing some additional literature, as well as adding an additional paragraph briefly discussing existing scholarship around how STCS can shed light on policymaking more broadly.

In addition, though the authors mention other reviews have been conducted, it would be worthwhile to expand on the general findings and more clearly articulate how this review is distinct.

RESPONSE:

We have briefly summarised some of the findings of the existing reviews where we introduce them (p.3, l.17), as well as expanding on the two reviews that focus on policy (Atkinson *et al.*) or draw inferences about policy (Langellier *et al.*) (p.5, l.1) . We have also added a statement in the 'aims and scope' section around how this review differs from existing systematic reviews on STCS in public health (p.6, l.1).

At a methodological level, it is not entirely clear why the review focuses on application to NCD prevention policies and whether there is sufficient literature to warrant a review. There are particular reasons why the policy making process for NCD prevention policies are distinct from other public health policies so there can be a justification for this scope but this is not clearly articulated in the current manuscript.

RESPONSE:

Thank you for this comment. As the review has progressed, we have identified over 100 relevant articles to include which described STCS-informed methods used in policymaking for NCD prevention. This is a rapidly growing field and there is a surprising amount of literature, much of which has been published in the last few years. We have added some detail around our focus on NCD prevention policy in the 'aims and scope' section (p.5, l.24):

STCS methods may be usefully applied in other areas of public health characterised by complex interactions between multiple stakeholders and domains, as well as other disciplines more broadly. However, policymaking for NCD prevention has characteristics which may make STCS methods particularly useful. First, there are commercial actors and interests involved in NCD-related policy, including the tobacco, alcohol, and food and beverage industries. This adds additional complexity to the policy process, and makes a case for transparent approaches to incorporating multiple perspectives and forms of evidence in making policy decisions. Second, despite concerted policy efforts to reduce the burden of NCDs, in many contexts progress has been limited, suggesting that a novel approach that encompasses complexity may be useful (17,18).

Finally, it would be helpful for the authors to include some indication of the impact of the the limitation of including only studies that self-define as STCS and what that definition may look like. The authors state that they "hope this will not result in the exclusion of a large number of relevant studies" but do not indicate steps to ameliorate this risk should the criteria end up excluding a large number of studies.

RESPONSE:

Thank you for this comment. We have now added some detail around this limitation and described the steps we will take to reduce the risk of relevant studies being included:

Further, studies that do not 'self-define' as using methods informed by STCS will not be included. Anzola and colleagues highlight the existence of 'analogical' uses of terms relating to complexity, where central characteristics of STCS are employed or implicitly referred to without being explicitly linked to the relevant theory and methods (31). Narrative reviews have previously identified implicit complexity concepts in the policy literature (8). However, in the absence of shared terminology, such usage may be difficult to systematically identify in the literature given the reliance of the systematic review method on identifying key words and phrases. In order to include as many relevant examples as possible, we will conduct title-abstract screening in an inclusive way, progressing records to full-text screening if there is any uncertainty. Further, our search strategy has been designed to be relatively inclusive, including broad terms related to STCS, such as complexity and system lens or perspective, as well as specific methods previously identified in systematic reviews.

Reviewer: 2

Dr. Benjamin Rigby, Durham University

Comments to the Author:

Thank you for the opportunity to review this manuscript. Please see the attached document for my full review.

Abstract

A couple of minor points:

1. Page 1 (line 18): Consider stating 'scoping' review in the aim of the study.

RESPONSE:

Many thanks, this has been added.

2. Page 2 (lines 7-8): 'This scoping review...' This reads like the aim of the study and thus duplicates your statement at the end of the introduction section of the abstract. Please remove this sentence and amalgamate with the previous statement.

RESPONSE:

Thanks for the suggestion. The synthesis section of the abstract has been updated to read:
Synthesis: We will identify relevant studies, summarising key data from each study and mapping applications of methods informed by STCS to different parts of the policy process. Review findings will provide a useful reference for policymakers, outlining which domains of the policy process different methods have been applied to.

Introduction

The manuscript will benefit from additional consideration of research that has previously explored applications of STCS, particularly theoretically, in the context of policy processes. I have provided some references below that the authors may wish to consider to help inform these considerations.

3. Page 3 (lines 7-9): Please reference this statement (i.e. STCS may be of value for those with limited time...)

RESPONSE:

Thank you for this comment! We have removed these statements at the end of the first paragraph as being more relevant to practical guidance we are working on around this topic than to the academic manuscript.

4. Page 4-5 (Applying STCS to the policy process): When reading this section, I was searching for a definition of the policy process. I found it in the methods section (page 7, line 21; table 1). Please bring this definition and table forward or summarise some key elements of policy processes in your introduction to aid the reader. It would be beneficial to state that this is just one conception of the policy process, and that while some policy scholars may denounce such linear/cyclical models, they are familiar to, and commonly considered by, policy actors.

RESPONSE:

Thank you for this suggestion. We have acknowledged that scholars have critiqued these types of models, but that they may be useful to a policy and practice audience. We have also moved the text introducing the CDC model of the policy process we are using, as well as the table with the domains, to this section of the introduction.

5. Page 4-5 (Applying STCS to the policy process): This section would also benefit from an acknowledgement of work that has looked at how theories of STCS may map on to aspects of the policy process, even if just to say these matters have been considered more theoretically, and less so in practical terms. In particular, please see Robert Geyer's (Lancaster University) or more pertinently Paul Cairney's (Stirling University), work, which offers insights on how complexity theories may inform policy-making and policy-making processes, taking into account policy-makers' perspectives and working practices.

RESPONSE:

Thank you for this helpful comment and the references provided below. We have added a paragraph briefly discussing some of these considerations and referencing Cairney and Geyer's work. We have also positioned the paragraph around the potential of STCS methods to facilitate the inclusion of more research evidence in policymaking after this paragraph as there are some related issues.

Methods and analysis

6. Page 7 (line 10): The second sub-research question is not clear. Do you mean to ascertain which methods mandate particular needs and levels of resource for policy-makers to use? Alternatively, do you mean which methods have been applied by policymakers according to their needs and the availability of resources to them? Furthermore, if the latter of these options, how do you intend to ascertain the level of resources policy-makers had access to; what sorts of resources will you consider charting from the data you retrieve (financial, human capital etc.)? Please clarify.

RESPONSE:

As the reviewer suggests, extracting details about the level of resource required for different methods from included papers has proved challenging. In piloting the review process, we therefore revised the research questions to focus on different 'practical considerations' around using STCS methods in the policy process. We have updated the protocol with these revised questions:

Informed by this aim, our central research questions are:

- How have methods informed by STCS been applied in the policy process in NCD prevention?
- Which domains of the policy process and areas of NCD prevention policy have methods informed by STCS been applied to?
- What practical considerations, such as advantages, limitations, barriers and facilitators, have been described in applying STCS-informed methods to NCD prevention policy?

7. Page 7 (lines 11-18): Again, these useful definitions may aid the reader if included earlier in the introduction instead. However, I am happy to leave this to the author's discretion.

RESPONSE:

We have moved these to the introduction as suggested.

8. Page 8: (line 3): How did you develop your search strings? Please make reference to said search strings and the Appendix in this subsection of the methods.

RESPONSE:

Detailed search strings in supplementary file 1 have now been referenced. Search string development is now briefly described:

Search strategies will be developed iteratively, informed by existing systematic reviews focused on related concepts (5,24–27) and indicator papers meeting inclusion criteria of which the authors are aware. As initial searches generated numerous records relating to genetics (due to the inclusion of the term 'regulation' along with health-related terms), a block of genetics-related NOT terms will be added to the search strategy.

9. Page 9 (lines 6 and 11): The focus of your scoping review is clear (i.e. STCS and NCD). However, what limits are you putting in place (i.e. do you have a pre-established set of STCS approaches/ NCDs you will consider)? I acknowledge STCS will be self-identified. However, this will then be subjective decision on the reviewers' part to include or exclude. While I am familiar with the authors' previous work in the field, others may not be. Please include a brief statement about the reviewers' requisite experience for undertaking this work. It will also be beneficial to offer a statement and/or examples of the kinds of STCS approaches and NCDs you expect to find data on.

RESPONSE:

Thank you for this comment. As this scoping review is exploratory, we were relatively inclusive in both our search strategy and our screening process. We did however make some decisions around the limits of the review. We have added more detail around this to this section of the methods:

For academic records, titles and abstracts will initially be screened, followed by full-text screening. Full-text screening will be undertaken by two independent researchers, one of whom will have extensive experience in the area of STCS and NCD prevention (TLP or EM, who have previously authored reviews on related topics (27,31)). In order to facilitate the identification of methods which may not have been identified as STCS methods by previous reviews of the public health literature (2,7,27), but which authors have identified as STCS methods, title-abstract screening will take an inclusive approach, and full texts will be screened to identify STCS language used to describe methods. Papers which focus on healthcare or clinical services rather than primary prevention will be excluded. Papers which concern potential risk factors for NCDs but focus on non-NCD outcomes (such as alcohol consumption as a risk factor for road traffic accidents or inter-personal violence) will also be excluded.

10. Page 11 (line 5): This short statement alludes to Arksey and O'Malley's optional consultation phase of scoping reviews. Consider changing the subtitle to reflect this stage, as set out by Arksey

and O'Malley, and briefly justify why you decided not to undertake it, particularly alongside policy-makers themselves.

RESPONSE:

Thank you for this comment. This subtitle must be left in place as it is required by BMJ Open publishing guidelines, and per the guidelines refers strictly to patients and members of the public, however it does not refer to policymakers. We did receive input around the scope and approach to this review from the WHO-Europe Office for NCD prevention, which we have noted in the acknowledgements.

Strengths and weaknesses:

11. Page 10 (lines 20-22): I was previously searching for a statement about why you were not including grey literature in the 'study selection' section on page 8. I do not see consider this much of a limitation as, a few key government reports aside (e.g. Magenta Book 2020), I do not anticipate much grey literature on the topic of interest. I recommend moving this statement to the methods section.

RESPONSE:

We have moved this statement to the methods section (Stage 2: Identifying relevant studies).

Ethics and dissemination

12. Ethics and dissemination are, largely, separate considerations. I recommend using two separate subheadings, both on pages 11-12 and in the abstract.

RESPONSE:

Thank you for this comment. We agree that these are distinct, but have included them jointly in line with BMJ Open submission guidelines for protocols (<https://bmjopen.bmj.com/pages/authors/#protocol>).

Appendix

13. Please number your Appendix (i.e. Appendix 1). Check the publication guidelines; this may need to be termed and referenced 'supplementary material/file' both here and in the main manuscript.

RESPONSE:

The appendix has been re-named Supplementary File 1 and is now referenced in the text as suggested.

PRISMA-P 2015 Checklist

14. Please double-check page numbers. There seem to be discrepancies between those stated in the checklist and where the information is found in the manuscript (e.g. funding information is on page 13, while the checklist says page 16).

Thank you, the checklist has been updated to match the revised manuscript.

General comments and reflections

15. Please be aware your page numbering seems to go wrong from page 10 onward (however, this may be a proof-formatting error).

RESPONSE:

Thank you for this comment. We believe this to be a proofing error as page numbers in our submitted file look in order.

Reference is made to Anzola's taxonomy of terminology associated with STCS. The authors may choose to reflect on this in the write-up of their review, to consider how policy-makers themselves discuss such concepts and thus may likely engage with the research findings.

RESPONSE:

Thank you for this suggestion.

Please be advised that new PRISMA reporting standards were published this year, and I recommend the authors reflect on these in their write-up.

RESPONSE:

Thank you for this comment. While the new PRISMA statement recommends the continued use of PRISMA-P 2015 for protocols, we will ensure that the full review is reported in line with PRISMA 2020.

Concluding remarks

I believe this scoping review will make a valuable contribution to the field of STCS and policy. I encourage the authors to consider and respond to my suggested revisions above.

RESPONSE:

Thank you very much for this thoughtful and detailed response to our manuscript!

I offer the following references that may assist in this regard:

- Cairney. (2012). Complexity theory in political science and public policy. *Political Studies Review*. 10(3), pp.346-358.
- Cairney. (2015). How can policy theory have an impact on policy-making? The role of theory-led academic-practitioner discussions. *Teaching Public Administration*. 33(1), pp.22-39.
- Cairney and Geyer. (2017). A critical discussion of complexity theory: how does 'complexity thinking' improve our understanding of politics and policymaking? *Complexity, Governance and Networks*. 3(2), pp.1-11.
- Nobles et al. 2021. The action scales model: a conceptual tool to identify key points for action within complex adaptive systems. *Perspectives in Public Health*. doi:10.1177/17579139211006747.
- Page et al. 2021. The PRISMA 2020 statement: an updated guideline for reporting systematic reviews. *BMJ*. 372(n71).
- Weible et al. eds. 2017. *Theories of the Policy Process*. 4th ed.

VERSION 2 – REVIEW

REVIEWER	Rigby, Benjamin Durham University
REVIEW RETURNED	19-Jul-2021

GENERAL COMMENTS	Peer Review Re. Systems thinking and complexity science methods and the policy process in non-communicable disease prevention: a systematic scoping review protocol Thank you for resubmitting a revised copy of the above-entitled manuscript. I welcome the opportunity to re-review it. I am satisfied that the authors have considered the reviewers' comments in suitable detail. In particular, the introduction better considers the existing evidence relating to these topics of focus. The rationale is also clearer.
--

	The manuscript is well presented and provides sufficient detail in its methods to enable future replicability if required. I recommend a final thorough proof-read of the manuscript before publication, as there were some instances, particularly in the revised sections, where more effective use of grammar or punctuation would aid the reader. It is typical for the authors to state in the contributions section that all authors read and approved the final manuscript.
--	--